

# Comprehensive analysis of the start-up period of a full-scale drinking water biofilter provides guidance for optimization

Loren Ramsay[1], Inês L. Breda[1,2], Ditte A. Søborg[1]

[1]Research Group for Energy and Environment, VIA University College, Horsens, 8700, Denmark.

[2]Department of Chemistry and Bioscience, Aalborg University, Aalborg, 9220, Denmark.

*Correspondence to*: Ditte A. Søborg (dans@via.dk)

**Abstract.** The use of biofilters to produce drinking water from anaerobic groundwater is widespread in some European countries. A major disadvantage of biofilters is the long start-up period required for virgin filter medium to become fully functional. Although individual aspects of biofilter start-up have previously been investigated, no comprehensive study in full-scale using inherent inoculation has previously been documented. A thorough investigation of a full-scale drinking water biofilter was carried out over 10 weeks of start-up. The many spatial and temporal changes taking place during start-up were

documented using a holistic approach. In addition to collection of many samples over time (frequency) and space (filter depth), this study entailed the use of multiple sample media (water, backwash water and filter media) and multiple types of analyses (physical, chemical and microbiological). The decrease in filter effluent concentrations of individual substances to compliance levels followed a specific order that was shown to coincide with the spatial-temporal development of bacteria on the filter media. Due to the abiotic nature of the iron removal process, iron disappears first followed by substances that

require growth of microorganisms: ammonium, with nitrite appearing briefly near the end of ammonium removal, then manganese. The thorough overall picture obtained by these efforts provides guidance for optimization and monitoring of the start-up. Guidance include to shorten the start-up by focusing on kick-start of the ammonium removal, to limit the monitoring burden to at-line measurements of ammonium in finished water samples supplemented with manual manganese measurements when ammonium removal is complete, and to improve filter design by isolating the removal processes in

separate, smaller filters.

**Keywords**: start-up; drinking water; rapid biofilter; iron; ammonium; manganese; full-scale; inherent inoculation; monitoring.



## 1 Introduction

Biofilters for the production of drinking water from anaerobic groundwater have been in widespread use for decades in some European countries and are currently gaining interest in North America. In this paper, biofilters are defined as submerged, granular, rapid filters in which the individual grains of the filter media have developed a natural coating capable of performing the desired treatment. A typical waterworks may be composed of an aeration step followed by biofiltration. This simple treatment train is adequate to remove iron, ammonium, manganese and other electron donors without the use of coagulants. The resulting drinking water is generally free of pathogens and biologically stable, making it possible to produce and distribute drinking water without any form of disinfection.

A start-up process is necessary for virgin filter media to become fully functional, capable of producing water that complies with drinking water criteria. In practice, commissioning of a new biofilter often includes four steps: 1) initial disinfection, 2) exaggerated backwashing to remove fines from the filter media, 3) inoculation of the filter medium with microorganisms and 4) formation of an inorganic coating and biofilm on the individual grains of the filter medium.

The start-up period is time-consuming, delaying the production of drinking water to the consumer. This delay causes a number of other disadvantages: raw water must be abstracted from a groundwater resource that may already be fully exploited, partially treated water must be discharged to a surface water recipient causing quality or quantity issues, energy must be used for pumping raw water and backwashing the filters, and an alternative source of drinking water must be made available to the consumers for the interim.

To shorten the start-up time, several studies have focused on inoculation of the biofilter using discrete (single or multiple) additions of a concentrated source of microorganisms. These methods may be termed "proactive inoculation" and are sometimes referred to as "rapid" or "accelerated" start-up. Methods include emplacement of mature filter media from a neighboring waterworks in the new biofilter (Qin et al. 2009; Zeng et al. 2010), dosing of backwash sludge from a neighboring waterworks (Štembal et al. 2004; Cai et al. 2015), and dosing of microorganisms isolated and grown in the laboratory (Li et al. 2005; Qin et al. 2009; Cai et al. 2015).

Rather than relying on proactive inoculation, operators may effectuate start-up using "inherent inoculation". This is simply the exposure of the filter media during start-up to microorganisms present in the raw water and in the water used for backwashing. Inherent inoculation microorganisms may stem from the indigenous aquifer microbial community, from biofilm in installations such as the well, riser pipe and raw water pipeline or from a mature filter of a neighboring waterworks that was used to produce the water used to backwash the new filter. This "natural" start-up method is widespread since it requires no extra inoculation activities. Further, it may be attractive to avoid contaminating virgin filter medium with mature medium that may contain invertebrates and the like.

Previous studies have shown that the time required for a biofilter to reach compliance is usually a few months when using inherent inoculation (Štembal et al. 2004). Possible reasons for this lengthy start-up period include low water temperatures, low availability of nutrients, weak inoculation or other factors (de Vet et al. 2012). A comparison of the length of the start-up



period from various studies in the literature is virtually futile, largely due to diverging experimental conditions and focus. For example, a long start-up period of around six months was found at unusually cold temperatures of 3-4 °C (Cai et al. 2016). In other studies, not all relevant electron donors such as ammonium (Zeng et al. 2010), manganese (Lytle et al. 2007) or iron (Bruins et al. 2015b) were included. Although individual aspects of biofilter start-up have previously been

investigated, no comprehensive study using inherent inoculation in full-scale has previously been documented.

Homogeneous, heterogeneous or biological processes (van Beek et al. 2012) cause the formation of a coating. Studies found that coated quartz sand had 10-55 times the iron(II) adsorption capacity of virgin quartz sand (Sharma et al. 2002). During start-up, iron oxide coatings are mainly formed by homogenous and heterogeneous iron removal. As filter media matures, biological oxidation may contribute to iron removal. An initial manganese oxide coating formed from bacteriological

manganese removal may be essential for the heterogeneous removal of manganese. Although debated, once an initial coating is formed, further removal of manganese is most likely dominated by physico-chemical processes due to the adsorptive and autocatalytic oxidative properties of the manganese oxide (Bruins et al. 2015a).

In contrast to iron and manganese, ammonium removal relies solely on bacterial processes. Formation of biofilm on filter media grains involves a critical initial cell attachment, which is influenced by a multitude of factors including surface charge,

hydrophobicity and surface roughness (Palmer et al. 2007). Following attachment, microorganisms proliferate and produce a matrix of extracellular polymeric substances. During attachment and growth, inoculated microorganisms that can adapt to the new aerobic conditions of the aerated raw water are naturally selected and a microbial community structure evolves in the filter medium. Variations in this microbial community in biofilters between waterworks may be explained by the groundwater chemistry of the raw water (Albers et al. 2015). One study identified differences in the microbial communities

in raw water, outlet water from pre-filters and outlet water from after-filters (Gülay et al. 2014). This indicates that a wide variety of inherent microorganisms is present in the raw water inoculum, but that the composition of the inlet water drives bacteria selection and growth. Bacterial groups that are involved in removal of ammonium, nitrite and manganese need to proliferate in the microbial community in order for the start-up to progress to compliance (Cai et al. 2015, Bruins et al. 2015b).

Monitoring of waterworks performance during start-up as well as during subsequent normal operation ranges from manual collection and analyses of water samples to wireless real-time sensor solutions. Physical, chemical or microbiological parameters are measured according to the need in a given situation and the cost for the waterworks. Monitoring during an intensive period such as start-up can therefore be expensive. Thus, knowledge of when, where, what and how to measure can have a great impact on the monitoring cost of a waterworks.

The aim of this study was to obtain a clear overall picture of the many spatial and temporal changes that take place during start-up using a holistic approach. The study investigated the typical start-up processes of a full-scale drinking water biofilter with inherent inoculation by measuring physical, chemical and microbiological changes in water, filter media and backwash samples. With the detailed documentation of interrelations of these parameters during the start-up processes, the study was able to provide guidance for the optimization of the start-up of biofilters.



## 2 Methodology

### 2.1 Design of the treatment process

The investigations described in this paper were carried out at Truelsbjerg waterworks near Aarhus, Denmark. This facility is composed of a well field with eight wells screened in an anaerobic aquifer and a treatment facility with three identical

production lines. The work described here is focused solely on one of these production lines.

Treatment starts with the addition of > 90 % pure oxygen to the raw water. The oxygen is produced on-site by oxygen generators using pressure swing adsorption technology (Oxymat A/S, Denmark). Immediately following oxygenation (less than one minute residence time), the water passes to two pressure filters placed in series, Filter 1 and 2 (Fig. 1).

Each filter has a cross-sectional area of 8.95 $m^2$ and a bed height of 2.29 m excluding the gravel support layers. A series of

13 water taps (ball valve and L-shaped 8 mm stainless steel tap that extends 30 cm into the filter media) and 13 filter media sampling ports (ball valve giving direct access to the filter media) at 20 cm depth intervals on each filter allowed collection of samples at different depths. The top three sampling locations access standing water above the filter media. Filter 1 is composed of granular calcium carbonate, and Filter 2 of granular quartz sand with a 20 cm layer of manganese oxide. Two different grain sizes of quartz gravel are used in each filter as a support material (Fig. 1). Virgin filter media was analyzed

for particle density and porosity by gravimetric methods. Grain size was determined by a photometric particle analyzer (Camsizer®2006, Retsch Technology GmbH, Germany). As reported in previous studies, the grain size ranges (10–90 % fractiles) were 2.3–4.1, 0.5–0.8 and 1.6–3.2 mm, the grain densities were 2.4, 2.5 and 3.3 kg $L^{-1}$, and the bed porosities were 40, 33 and 44 % for calcium carbonate, quartz sand and manganese oxide, respectively (Søborg et al. 2015; Breda et al. 2016).

### 2.2 Raw water

The raw water is composed of a mixture of anaerobic groundwater abstracted from several wells at a time, reflecting variations in hourly consumer demands and planned well rotation. Table 1 shows the average raw water quality measured in unfiltered samples collected during the start-up period.

In general, the raw water is saturated with calcium carbonate, the pH conditions are near neutral and there is a very low

concentration of organic carbon (measured as non-volatile organic carbon, NVOC). Redox conditions prior to oxygenation are reduced (no oxygen or nitrate, but traces of methane and hydrogen sulfide). During the start-up period, the oxygen content of the water entering Filter 1 was generally around 5 mg $L^{-1}$. The water temperature remained constant at around 9 °C. Speciation calculations using the aqueous geochemical model PHREEQC (Parkhurst and Appelo, 2013) showed that the dissolved iron and manganese in the raw water are dominated by free divalent cations with various aqueous complexes as

minor components.



### 2.3 Filter operation

Initial disinfection of the treatment system following emplacement of new filter media was carried out with a commercial mixture of acid and hydrogen peroxide that was recirculated for 24 hours, followed by a double backwash. Inherent inoculation was used during start-up by introducing microorganisms from two sources: the groundwater wells used as the source of raw water and unchlorinated drinking water from a nearby waterworks used for backwashing.

Day 0 represents the first day with normal operation of the filters. During start-up approximately one-third of the intended production flow was used (with a slight increase around Day 33). The start-up investigation concluded on Day 70. The filters were backwashed frequently (daily in Filter 1, then Filter 2) the first eight days of operation and thereafter every four days. The water used for this backwashing process stemmed from a nearby well-functioning waterworks and was therefore composed of pure drinking water with no chlorine, iron, manganese or ammonium. Throughout the start-up period, the filter backwash was based on a three-step procedure as follows: 3 minutes with compressed air alone (60 $m^3$ $m^{-2}$ $h^{-1}$) followed by 10 minutes with air (60 $m^3$ $m^{-2}$ $h^{-1}$) and water (11 $m^3$ $m^{-2}$ $h^{-1}$) simultaneously followed by 9 minutes with water alone (29 $m^3$ $m^{-2}$ $h^{-1}$).

### 2.4 Sampling of water and filter media

Unfiltered water samples were collected using standard procedures (International Organization for Standardization, 2006) from stainless steel taps at 29 locations: from raw water, water between filters, finished water and the 13 water taps on Filter 1 and 2 (Fig. 1). All samples were stored at 5 °C and analyzed within 24 hours, except samples for qPCR, which were filtered and stored at -21 °C until analysis.

Filter media samples (~1 kg) were collected from the sampling ports on each filter under operating conditions by inserting a 1 m long hollow stainless steel probe horizontally. Filter media samples were collected along the entire length of the probe as it was inserted into the filter media ports. Filter 1 was sampled at filter media depths 10, 50, 90 and 190 cm (Fig. 1, ports 4, 6, 8 and 13) while Filter 2 was sampled at depths 10, 70, 130 and 190 cm (Fig. 1, ports 4, 7, 10 and 13). Filter media samples were stored at 5 °C, drained and divided using the two-dimensional Japanese slab cake method (Pitard 1993) prior to analyses. DNA samples for qPCR were extracted within 24 hours and stored at -21 °C until analysis.

Composite samples (~4 L) of backwash effluent from Filter 1 and 2 were collected manually during the course of six backwash events (Days 0, 10, 34, 38, 42 and 66). Each composite sample was prepared by combining 100-300 mL increments of unfiltered backwash water in a flow proportional manner, depending on the backwash flow. Increments were collected once a minute.

### 2.5 Analyses

At-line/in-line measurements were carried out at several locations. Seven parameters often measured at Danish waterworks were included: flow, pressure, dissolved oxygen, pH, conductivity, temperature and turbidity. In addition, at-line ammonium





concentrations were measured spectrophotometrically in triplicate every 40 minutes during the start-up period using Amtax SC ammonium analyzer (Hach, USA).

Physical tests included pressure profiles which were registered manually by inserting a digital manometer (LEO Record, KELLER AG, Switzerland) in the 13 filter media ports on each filter (Fig. 1). Values were corrected for the height

difference of the ports. Turbidity and suspended solids were measured in backwash samples and reported previously (Breda et al. 2016).

Chemical analysis of water samples included the substances iron, manganese and ammonium which were analyzed according to the manufacturer's instructions with the standard kits LCK320/321/521, LCK304, and LCW532, respectively, using a spectrophotometer (DR 3900, Hach, USA). Concentration of nitrite was analyzed by Standard Methods 4500-$NO_2$

(B) (APHA/AWWA/WEF, 1989). Samples for iron and manganese were not filtered as tests showed that virtually all iron and manganese passed through 0.45 µm filters (data not shown). Additional standard water quality parameters (Table 1) were analyzed by Eurofins, Denmark. Composite backwash samples were analyzed for iron, manganese and calcium fines by Standard Methods 3120 (APHA/AWWA/WEF, 1989) following acid digestion by Eurofins, Danmark.

Microbiological analyses were performed by Eurofins, Denmark and included measurements of four bacterial counts; colony

forming units (CFU) at 22 °C, CFU at 37 °C, coliforms and *E. coli* as required in the national drinking water regulations. Further, ATP of 1 g filter media collected from depths 10, 50, 130 and 190 cm from Filter 1 and 2 was measured using the deposit and surface analysis (DSA) kit and the PhotonMaster™ (both LuminUltra Technologies Ltd., Canada) according to the manufacturer's instructions.

DNA was extracted in triplicate from 200 mg filter media samples using the PowerBiofilm® DNA Isolation Kit (MoBio

Laboratories Inc., USA) according to the manufacturer's instructions. DNA concentration and purity was evaluated spectrophotometrically by determining the UV absorption at 260 nm (A260) and the A260/A280 ratio, respectively, by using a NanoDrop™ 1000 Spectrophotometer (NanoDrop Technologies, USA). The two extractions with the highest DNA amount and purity were analyzed using qPCR.

Quantitative analysis using qPCR was performed for total bacteria (Eubacteria), total Archaea, ammonium oxidizing bacteria

(AOB), *Nitrospira*, *Nitrobacter* and *Leptothrix* with group specific 16S rRNA gene primers (modified from Degrange and Bardin 1995; Ferris et al. 1996; Siering and Ghiorse 1997; Hermansson and Lindgren 2001; Graham et al. 2007; Porat et al. 2010), while quantification of ammonium oxidizing archaea (AOA) was performed with group-specific primers (modified from Francis et al. 2005) targeting the functional *amoA* gene coding for the enzyme ammonium monooxygenase. PCR was performed with 25 µl reaction mixtures containing 12.5 µl of 2×iQ SYBR Green Supermix (Bio-Rad Laboratories, Inc.,

Denmark), 500 nM of each primer, 10 ng template DNA and DNA/RNAse free water to 25 µl. PCR was performed with a MX3000 Real Time PCR system (Stratagene, USA) using optimized programs based on the mentioned references.

Planktonic bacteria were measured at-line every 10 minutes throughout the start-up period in finished water with a bacterial monitor (GRUNDFOS BACMON, Grundfos Holding A/S, Denmark). This optical sensor system utilizes a light source, a



camera arrangement and an image analysis system to quantify individual particles in a flow cell (Hojris et al. 2016). The particles are classified as bacteria or abiotic particles.

## 3 Results and Discussion

### 3.1 Removal of fines in de initial frequent backwashes

Iron was virtually absent in Filter 1 backwash water the first four days whereafter concentrations increased rapidly (Fig. 2). This indicates that no fines containing iron were present in the calcium carbonate medium and that iron in the raw water passed through Filter 1 or was accumulated in a form that could not be removed by backwashing during the first four days. The subsequent increase of iron in backwash water of Filter 1 shows that iron accumulated in the filter medium was removed by backwashing after Day 4. For Filter 2, the decreasing curve for iron indicates that fines containing iron were removed

from the quartz and/or manganese oxide media during the first 4 days of commissioning (Fig. 2).

Manganese was removed with the backwash water from Filter 2 during the initial backwashes. Initially, composite backwash samples from Filter 2 showed up to 4 mg $L^{-1}$, whereafter values dropped to below 1 mg $L^{-1}$ in a couple of days (Fig. 2). The higher initial values were likely caused by manganese oxide fines present in the virgin manganese oxide filter medium. Virtually no manganese was removed with backwash water from Filter 1.

The majority of the total calcium in the backwash water (Fig. 2) stems from dissolved calcium present in the treated water that was used for backwashing. However, backwash water from Filter 1 in general contained slightly more total calcium than backwash water from Filter 2 which can be explained by fines present in the virgin granular calcium carbonate filter medium in Filter 1. This was corroborated by a milky appearance of the backwash water from Filter 1 during the first days of commissioning.

Different types of filter media may present different risks for increased turbidity in the treated water or cementing of the filter media. In general, five backwashes (initial double backwash plus three daily backwashes) were sufficient to remove fines present in the three different types of virgin filter media included in this study.

### 3.2 Flow, pressure and residence time distribution

Figure 3 shows the flow and the pressure drop over Filter 1 for the entire 70 days start-up period. The repeating pattern

(every four days) was caused by the backwash frequency. As each filter run progressed, the pressure drop (black) increased, mirrored by a decrease in the flow (grey) (Fig. 3). It was noted that each backwash was successful in restoring the pressure and flow to initial values. During the first 40 days of the start-up period, the pressure drop increased more rapidly and to higher values during each filter run, indicating a progression in the clogging phenomenon.

A series of pressure measurements carried out at various depths in Filter 1 at the end of filter runs just prior to backwashing

showed the change in pressure drop under low flow conditions occurred solely in the top 30 cm of the filter medium (Fig. 4). This indicates that clogging took place at this depth interval. On Day 2, a slightly higher pressure drop in general was caused



by a slightly higher flow. No change in the pressure drop was observed on Day 2 near the top of the filter medium, since this was in the period of short filter runs with the initial daily backwashes. The sudden pressure increase near the bottom of the filter on this day was unexplained. After the start-up period, the flow was increased about three times for production of drinking water. More than a year later (Day 515), it was shown that this flow increase caused a higher pressure in general

and a less pronounced pressure drop at the top of the filter (Fig. 4).

The average filtration rate during the start-up period was 5.0 $m^3\, m^{-2}\, h^{-1}$ (calculated from an average flow of 45 $m^3\, h^{-1}$, Fig. 3). Results of a previously reported tracer test showed that the majority of the residence time of water in Filter 1 occurred above and below the actual filter bed. The residence time of the water in contact with filter media was calculated to 12 minutes (Søborg et al. 2015).

### 3.3 Iron removal

An iron loading rate based on filter area was calculated to be 7.0 g Fe $h^{-1}\, m^{-2}$. The total iron load during a four-day filter run was therefore 0.7 kg Fe $m^{-2}$ filter area, which is within the range of the practical rule of thumb that backwash should be carried out after loading the filter with about 0.5-1 kg Fe $m^{-2}$.

Iron was removed completely (i.e. below the national criteria of 0.1 mg $L^{-1}$) in Filter 1 from the beginning of the start-up period (Fig. 5), suggesting homogenous and/or heterogeneous removal rather than biological. This immediate iron removal on the filter media supports other results (Štembal et al. 2004) and was expected with the neutral pH and low level of organic matter of the raw water (Table 1) which reduces the likelihood of Fe(III) dissolution and metal-humic acid complexation, respectively.

Due to a technical fault which caused the filters to be deprived of oxygen for a short period (Day 49-51), a small increase in iron concentration was seen in Filter 1 effluent on Day 51 (Fig. 5). The effects of oxygen deprivation were investigated in detail at the same waterworks in a subsequent study (Søborg et al. 2015).

Composite samples of the initial frequent backwashes of Filter 1 on Day 1-4 indicated that insignificant amounts of iron were removed during backwash at this early stage (Fig. 2). After the first week, > 80 % of iron in the raw water was removed

by backwashing, and steady state iron concentrations in backwash water were achieved within the interval Day 8-11 (Breda et al. 2016). This means that up to 20 % of the iron removed on Filter 1 accumulates in the filter, contributing to the desired inorganic coating on the media grains and the undesired increase of grain size and filter bed volume.

With the low flow conditions of the start-up period, iron was removed to compliance within the first 30 cm of the filter medium in Filter 1 (except Day 5). The depth profile of iron concentrations in Filter 1 (Fig. 5) shows a similar pattern to the

pressure drop profile (Fig. 4) and it is presumed that accumulation of iron oxides in this thin layer is the main reason for the clogging of the filter. Undesirable formation of iron oxide flocs via homogeneous iron removal, which in turn is caused by the long retention time in the standing water above the filter medium may explain why this clogging is limited to the top layer of the filter. If iron oxides accumulate in an increasingly narrower depth interval over time, this can explain the



observed progression of the pressure drop during the first 40 days (Fig. 3). Iron measurements with smaller depth increments and more frequent measurements within each filter run may confirm this.

## 3.4 Ammonium removal

An ammonium loading rate based on filter area was calculated to be 1.1 g $NH_4^+$ $h^{-1}$ $m^{-2}$. The continuous measurements of ammonium using the Amtax SC ammonium analyzer were valuable to follow the progress of start-up. The onset of ammonium removal occurred around Day 32. This long period characterized by constant ammonium concentration in the finished water supports previous findings in filters utilizing inherent inoculation (Frischherz, 1985) and support that ammonium removal is biological. Within 12 days (Day 44) following onset, ammonium removal was complete (i.e. below the national standard of 0.05 mg $L^{-1}$). During this period, a spike of nitrite, the product of the first step of nitrification, was observed (Fig. 6). The highest measured nitrite concentration was 0.36 mg $L^{-1}$, well above the national standard of 0.01 mg $L^{-1}$. Nitrite in finished water met drinking water criteria by Day 50. The concentrations of ammonium and nitrite in finished water exceeded standards between Day 49 and 52 (Fig. 6) due to the deprivation of oxygen from the aforementioned technical fault.

By the end of the start-up period (Day 67), ammonium was removed to compliance at 70 cm depth of Filter 1. The main ammonium removal (89 %) took place in the interval 30-70 cm, immediately below the iron removal stratum, with little overlap between the two strata. In this depth interval, the volumetric loading rate was 2.3 g $NH_4^+$ $m^{-3}$ filter medium $h^{-1}$ and the volumetric removal rate was approximately 2.0 g $NH_4^+$ $m^{-3}$ filter medium $h^{-1}$.

## 3.5 Manganese removal

A manganese loading rate based on filter area was calculated to be 2.3 g Mn $h^{-1}$ $m^{-2}$. The onset of manganese removal in Filter 1 was not seen until Day 42 (Fig. 5), suggesting that growth of manganese oxidizing bacteria was needed. Near the end of the start-up period (Day 67), 91 % of the manganese in the raw water was removed on Filter 1. At this time, compliance with the national standard of 0.02 mg $L^{-1}$ was achieved in Filter 2.

As seen for iron and ammonium, the unintentional oxygen deprivation on Day 49-51 also had a negative effect on the removal of manganese (Fig. 5). The manganese concentration in the effluent of Filter 1 increased to 1.56 mg $L^{-1}$ on Day 58, far above the raw water concentration, indicating the mobilization of manganese previously accumulated on the filter medium (Søborg et al. 2015).

Significant removal of manganese (from 0.46 to 0.07 mg $L^{-1}$) was observed in Filter 2 already in the first sample (Day 5). This manganese removal may be due to the presence of the manganese oxide filter medium and adsorption/autocatalytic oxidation processes (Bruins et al. 2015a) and shows one of the potential effects of selecting specialized filter media. Visual observation of media samples showed that the granular manganese oxide initially located on the top of Filter 2 sank through the quartz sand during backwash early in the start-up period due to its higher density.



On Day 67, manganese removal took place below a filter medium depth of 90 cm in Filter 1. The main manganese removal was seen in the interval 150-190 cm (Fig. 5) with no overlap with the ammonium removal strata. At this time, black particles presumably composed of manganese oxide precipitates were observed on the filter medium at the bottom of Filter 1. Compliance was reached at a filter medium depth of 10 cm in Filter 2. Compliance was observed at the bottom of Filter 1 in

a sample collected after more than one year of operation (Day 515, data not shown). This shows that Filter 2 was superfluous under the typical operating conditions. Some degree of overdesign of filter capacity is common in Denmark to allow for future water-production demands such as higher flows or higher inlet concentrations.

At the end of the start-up period, composite backwash samples as expected showed that the backwash procedure did not remove manganese oxides from the filter medium. Virtually all of the manganese that was removed from the raw water in

Filter 1 was accumulated on the filter media rather than being backwashed out (Breda et al. 2016). This shows that the manganese oxide coating on the filter media builds up year after year that the filter is in operation and is continually renewed, which may be important for the autocatalytic oxidation process.

In the most active 40 cm interval (150-190 cm), the volumetric manganese loading rate on Day 67 near the end of the start-up period was approximately 4.7 g Mn m$^{-3}$ filter medium h$^{-1}$. Since 74 % of the manganese entering this interval was

removed, the volumetric manganese removal rate was approximately 3.5 g Mn m$^{-3}$ filter medium h$^{-1}$.

### 3.6 Abundance of Eubacteria and Archaea in filter media

Total number of bacteria (Eubacteria) measured by qPCR was relatively constant throughout the start-up period in filter media samples collected from Filter 1. Bacteria relevant for the removal of iron and manganese (*Leptothrix*), ammonium (AOB) and nitrite (*Nitrospira*), however, increased considerably in number over time (Fig. 7). The most pronounced increase

in these bacteria took place between measurements made on Day 43 and 67 (the end of the start-up period). As nitrite and manganese did not reach compliance until after Day 43, an increase in *Nitrospira* and *Leptothrix* in that period was expected. In contrast, ammonium removal was almost complete on Day 43. The increase in AOB after this day indicated that a steady state had not yet been reached and that the microbial community was still developing.

Depth profiles for Eubacteria showed a maximum of $10^7 – 10^8$ copies g$^{-1}$ filter medium with the highest results at the top two

sampling depths (10 and 50 cm depths in Filter 1, Fig. 7). This observation was not further investigated, but possible explanations for the number of Eubacteria in the top of the filter may involve 1) the greater surface area due to precipitation of iron oxides at the top, 2) the continuous sorption of inoculation bacteria from the raw water and/or 3) bacterial growth due to presence of methane and other assimilable organic carbon (AOC) in the water which was then depleted at greater depths. The main source of Eubacteria on the filter medium is most likely sorption of inherent bacteria present in the raw water.

Only a small part of the total bacteria can be explained by the growth of specific autotrophic bacteria such as *Nitrospira* and *Leptothrix* (e.g. *Nitrospira* has been found to account for about 17 % of the total bacteria, Gulay et al. 2016) or by the growth of heterotrophic bacteria on AOC (assuming a yield of $10^6$ cells μg$^{-1}$ carbon, Prest et al. 2016).

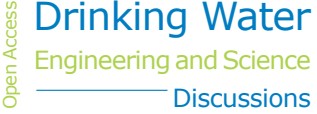

The biological stability of disinfectant-free drinking water (i.e. lack of aftergrowth in the distribution system following treatment at the waterworks) is largely determined by the amount of available nutrients. The growth limiting substrate in drinking water systems is often organic carbon (Prest et al. 2016). Measurements of methane on Day 39 showed removal of methane from 0.042 to 0.009 mg L$^{-1}$ between depths 10 and 50 cm on Filter 1, indicating that methane is used for bacterial growth. Measurements of NVOC throughout the start-up period showed an average of 1.6 mg L$^{-1}$, both in raw water and in Filter 1 effluent. Although no change in NVOC concentration was detectable with the method used, the AOC that typically accounts for few percent of the NVOC may play a role in bacterial growth in the top of Filter 1.

Measurements of ATP in filter medium samples indicated that the microbial activity was highest in the top of Filter 1. ATP content in the top half of Filter 1 ($10^2 - 10^3$ pg ATP g$^{-1}$ filter medium) was hundredfold higher than in the bottom half of Filter 1 and all of Filter 2. Hence, the highest numbers of total bacteria and the greatest activity was seen at the top of Filter 1.

At the end of the start-up period (Day 67), a high number of *Leptothrix* (~$10^6$ copies g$^{-1}$ filter medium) was found in the top layer of Filter 1 (10 cm, Fig. 7). This is in agreement with the removal of iron that took place in this vicinity (Fig. 5). Further, it showed that although initially dominated by chemical removal, biological oxidation of iron contributed to iron removal with time. Similar amounts of *Leptothrix* were found near the bottom of Filter 1 (190 cm, Fig. 7) which may be due to possible involvement of this genus in manganese removal (Bruins et al. 2015a; Cai et al. 2016). The highest amount of AOB (~$10^7$ copies g$^{-1}$ filter medium) was found at a depth of 50 cm (Fig. 7), in agreement with the depth interval at which ammonium removal took place (Fig. 5). Nitrite oxidizing bacteria were investigated by quantifying the genera *Nitrospira* and *Nitrobacter. Nitrospira* was below detection limits in many samples. The highest number ($10^5 - 10^6$ copies g$^{-1}$ filter medium) was found at 90 cm. This is likely because no nitrite substrate is present in the filter until the depth where ammonium begins to be oxidized. The number of *Nitrobacter* was below detection limits in all samples.

At the end of the start-up period, the numbers of total Archaea and AOA were relatively low (<$10^3$ copies g$^{-1}$ filter medium) and sometimes below the detection limit, indicating that AOA were less important than AOB for ammonium removal in this biofilter. In a few cases, the number of AOA exceeded the total Archaea by 50 %, likely due to the fact that AOA often carry 3 *amo* operons including *amoA* genes (Norton et al. 2002) whereas the archaeal genome normally has a single copy of the 16S rRNA gene (Vetrovsky and Baldrian, 2013). Nevertheless, this indicates that the majority of Archaea on the filter medium belonged to the AOA group.

The average amounts of bacterial groups were calculated over the entire depth of Filter 1, using a depth-weighted average. By the end of the start-up period, *Nitrospira, Leptothrix* and AOB accounted for approximately 1, 3 and 10 %, respectively, of Eubacteria. Despite their importance for water treatment, this shows that these autotrophs are not dominant in the biofilter at this stage in the microbial community development. Greater percentages of *Nitrospira* were found in previous studies of mature filter media (Gülay et al. 2016; Tatari et al. 2016).



### 3.7 Heterotrophic plate counts, coliforms and *E. coli*

HPC 22°C results ranged from 4-40 CFU mL$^{-1}$ in samples of raw water, water between filters and finished water (eight samples collected at various times during the start-up) and from 9-160 CFU mL$^{-1}$ in water samples collected at four depths from within each filter (Day 39 and 67). In the same samples, HPC 37°C results ranged from 1-10 CFU mL$^{-1}$ and 2-27 CFU

mL$^{-1}$, respectively. No coliforms or *E. coli* were detected in water samples at any location or time during the start-up period. The general low numbers of these bacteria showed that the entire chain of hygiene procedures used during well construction, waterworks construction, filter media transport and emplacement, initial disinfection of the treatment system, start-up processes, sampling and analyses was successful in avoiding contamination. By the end of the start-up (Day 67), results for all four bacteria measurements complied with national standards (Table 1) in finished water and did not cause delays in the

distribution of drinking water to the consumer.

### 3.8 Planktonic bacteria in finished water

In addition to traditional monitoring methods, the automated bacterial monitor (GRUNDFOS BACMON) was included (Hojris et al. 2016). In the period from the end of the initial daily backwashes (Day 8) until ammonium removal reached compliance (Day 44), concentrations of bacteria and abiotic particles were constant in the finished water. In general, the total

bacteria (black) and abiotic particles (grey) (Fig. 8) were in the range of 1000-3000 counts mL$^{-1}$ which is relatively low in comparison with other Danish waterworks (Hojris et al. 2016). It is interesting to note that this level was reached already two weeks after the beginning of the start-up period. The pattern of backwashes is seen every four days as short-lived spikes of high counts. This is an expected pattern, since up to an hour may be required before the filters effectively capture or wash out the particles that were suspended during the backwash. In the last third of the start-up period (approximately Day 45-67),

a distinct pattern of three peaks lasting several days each was observed. It was noted that the bacteria/abiotic particles ratio increased in the first two peaks, while it decreased in the third peak (Fig. 8). The changes observed at the same time as these peaks which may influence this pattern included high nitrite concentrations (peak 1), lack of oxygenation (peak 2) and hydraulic irregularities (peak 3). Further investigation would be required to determine if these changes are the direct cause of the peaks.

Monitoring regulations do not require high-frequency at-line measurements of bacteria. However, these measurements can indicate whether the function of the biofilter is stabile prior to allowing the finished water to be sent to the consumer and can also reveal information that would otherwise have gone undetected. However, methods for measuring total bacteria are not of value for monitoring the growth of specific bacterial groups in drinking water system. For instance, no increase in total bacteria was observed between the onset of nitrification and total ammonium removal (Day 32 to Day 44).




### 3.9 Spatio-temporal overview of start-up

The start-up period for drinking water biofilters may be divided into five conceptual phases which provide convenient milestones to express the progress of the start-up (Fig. 9A). Although often entailing a less complete data set, previous experience with start-up at Danish waterworks (data not shown) indicate that these phases are typical for biofilters treating

anaerobic groundwater. In Phase 1, heterogeneous iron removal increases over time until compliance is reached, occurring abiotically through the sorption of dissolved iron(II) and oxidation by oxygen to iron oxides. This phase is often short and may be less than a day. As the filter matures, the mechanism for removal can change, with microorganisms playing a greater role (van Beek et al. 2012). In Phase 2, no significant changes in finished water concentrations are apparent. This phase often drags on for weeks, to the dismay of the treatment plant operators. Phase 3 begins with the onset of ammonium removal and

continues until ammonium concentrations drop to compliance. The presence of iron has previously been shown to inhibit nitrification (de Vet et al. 2009). A spike of nitrite often emerges in Phase 3 as the first step of nitrification converts ammonium into nitrite. This peak disappears in the subsequent Phase 4 as nitrite is converted to nitrate. Phase 5 begins with the onset of manganese removal and continues until manganese concentrations drop to compliance. Here, it appears that presence of nitrite can inhibit the oxidation of manganese (Vandenabeele et al. 1995). At the end of Phase 5, all constituents

are in compliance and the start-up period may be considered complete (assuming the microbiological quality of the finished water is acceptable). The duration of each conceptual phase in this study was 0, 32, 12, 6 and 16 days, respectively, totaling approximately 10 weeks.

The granular media in a biofilter may be divided into strata according to filter depth. The existence of these strata may be illuminated by the analysis of water and filter media samples collected at various depths throughout the filter. The degree of

separation or overlap of these strata likely depends on a number of factors including hydrodynamic dispersion in the longitudinal direction during normal operation as well as the mixing of filter media during backwash. In this study, little overlap in the most active parts of the removal strata was observed. By the end of the start-up period, the main strata of iron, ammonium and manganese removal were located at depth intervals of 0-30, 30-70 and 150-190 cm in Filter 1, respectively (Fig. 9B). Analysis of water from a follow-up sampling event on Day 515 indicated somewhat greater overlap, possibly due

to the higher flow and/or due mixing of the filter media in connection with the greater number of backwashes.

Findings of this study in combination with previous studies (e.g. Frischherz, 1985 - conceptual phases and Tatari, 2016 - stratification) suggest that the spatio-temporal nature of the start-up period is a general condition which applies to all biofilters treating anaerobic groundwater regardless of filter medium, raw water quality, etc.

### 3.10 Guidance for optimization and monitoring of start-up

Based on the main findings of this study, one may consider the following optimization and monitoring recommendations when following start-up of drinking water biofilters:



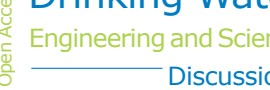

- The division of the start-up process into five conceptual phases showed that Phase 2 (which follows iron removal and preceeds ammonium removal) was by far the longest. Optimization of the start-up should therefore focus on kick-starting the ammonium removal.

- Stratification of the removal processes in the filter media during start-up has implications for pro-active inoculation. When inoculating with mature filter media, for instance, the inoculant should be collected from the proper depth interval of the donor waterworks and emplaced in the proper depth interval of the receiving waterworks. For practical reasons, it is therefore recommended to emplace the mature filter media during the initial filling of the filter.

- This study showed that manganese in the raw water can be removed by using manganese oxide filter medium. If the start-up process reaches Phase 5 but no further, addition of this medium may be considered to adequately remove manganese.

- This study has shown the value of various monitoring methods to describe the physical, chemical and microbiological changes during the start-up period. If the sole purpose is to monitor the progression of the start-up phases, however, the results from this study show that monitoring may be limited to at-line measurements of ammonium in finished water samples supplemented with manual manganese measurements after Phase 3 is complete.

- An in-depth analysis of the start-up period was possible in this study through the collection of samples at different depth intervals. From a practical point of view, these samples are unnecessary if start-up progresses as expected and if the goal is only to determine when compliance is reached. If there are difficulties during the start-up process, however, depth samples are valuable. Therefore, it is recommended to prepare biofilters during the construction phase with sampling ports at 10 or 20 cm depth intervals.

- Stratification in filters suggests that the various removal processes (for iron, ammonium and manganese) may be isolated in separate, smaller filters. This would allow for the tailoring of each process individually through pro-active inoculation, selection of filter media, biostimulation, flow, backwash and many other activities. Further studies are needed to confirm the effect of this re-design of the water treatment process.

## 4 Conclusions

A thorough investigation of 10 weeks of start-up was carried out using a holistic approach that documented the many spatial and temporal changes that take place during start-up. Traditional and advanced monitoring methods were used to analyze the physical, chemical and microbiological changes and their interrelation in water, filter media and backwash water. Results of this study provide guidance for optimization of the start-up to shorten the start-up period, to reduce the monitoring burden and to improve filter design.




**Data availability**

The research data of this work can be obtained by contacting the corresponding author.

**Competing interests**

The authors declare that they have no conflict of interest.

**Acknowledgements**

This project was financed by VIA University College and Aarhus Water A/S. The authors would like to thank Aarhus Water A/S, Silhorko-Eurowater A/S, Xergi A/S, Grundfos Holding A/S and Hach for assistance and technical expertise.

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

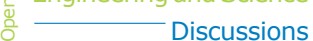

**Table 1: Raw water quality at Truelsbjerg waterworks.**

| Parameter | Unit | Average | Std. Dev. | Danish criteria[b] | No. of samples |
|---|---|---|---|---|---|
| **Treatment substances** | | | | | |
| Oxygen[a] | mgL⁻¹ | 0.22 | 0.12 | >5 | 9 |
| Iron | mgL⁻¹ | 1.40 | 0.19 | 0.1 | 26 |
| Manganese | mgL⁻¹ | 0.45 | 0.07 | 0.02 | 28 |
| Ammonium | mgL⁻¹ | 0.21 | 0.04 | 0.05 | 26 |
| Methane | mgL⁻¹ | 0.07 | 0.02 | 0.01 | 3 |
| **Major ions** | | | | | |
| Calcium | mgL⁻¹ | 89 | 5 | 200 | 4 |
| Magnesium | mgL⁻¹ | 7.6 | 0.3 | 50 | 3 |
| Sodium | mgL⁻¹ | 15 | 0.0 | 175 | 3 |
| Potassium | mgL⁻¹ | 2.2 | 0.0 | 10 | 3 |
| Hydrogen carbonate | mgL⁻¹ | 282 | 4 | >100 | 4 |
| Chloride | mgL⁻¹ | 25 | 1 | 250 | 9 |
| Sulfate | mgL⁻¹ | 38 | 5 | 250 | 9 |
| Nitrate (as $NO_3^-$) | mgL⁻¹ | <0.5 | 0.0 | 50 | 8 |
| **Bacteria** | | | | | |
| HPC, 22°C | mL⁻¹ | 17 | 11 | 50 | 8 |
| HPC, 37°C | mL⁻¹ | 1.9 | 1.5 | 5 | 8 |
| Coliforms | 100mL⁻¹ | <1 | 0.0 | <1 | 8 |
| *E. coli* | 100mL⁻¹ | <1 | 0.0 | <1 | 8 |
| **Others** | | | | | |
| Nitrite (as $NO_2^-$) | mgL⁻¹ | 0.01 | 0.01 | 0.01 | 17 |
| Phosphorous (as P) | mgL⁻¹ | 0.08 | 0.02 | 0.15 | 4 |
| NVOC (as C) | mgL⁻¹ | 1.6 | 0.1 | 4 | 9 |
| Hydrogen sulfide[c] | mgL⁻¹ | <0.02 | 0.00 | 0.05 | 3 |
| pH[a] | | 7.3 | 0.06 | 7.0-8.5 | 9 |
| Conductivity[a] | mSm⁻¹ | 58 | 0.8 | >30 | 9 |
| Temperature[a] | °C | 8.9 | 0.1 | 12 | 9 |

Shading indicates concentrations above Danish drinking water criteria.

[a] Field measurements.

[b] Criteria for water exiting the waterworks (BEK 802, 2016).

5     [c] Very faint hydrogen sulfide odor is detectable at the waterworks.



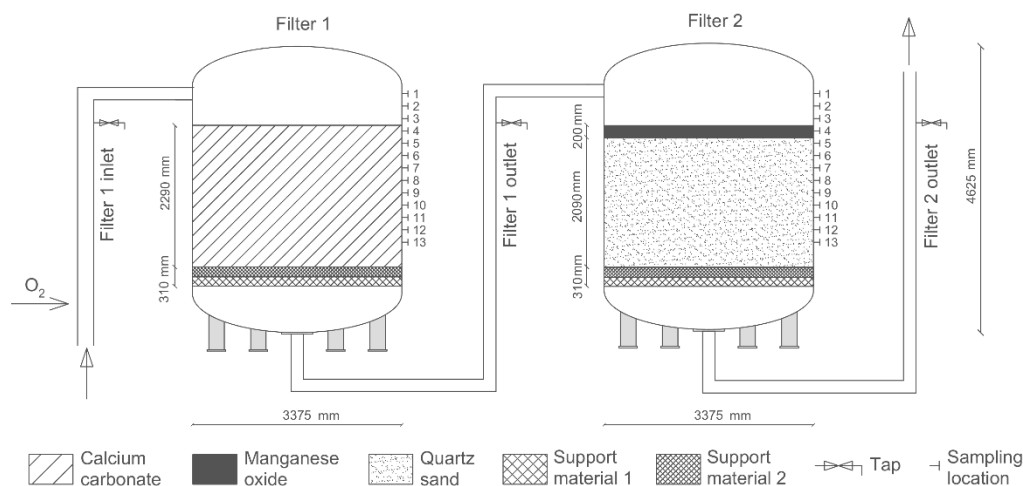

**Figure 1: Initial design of a production line at Truelsbjerg waterworks (revised from Søborg et al. 2015).**

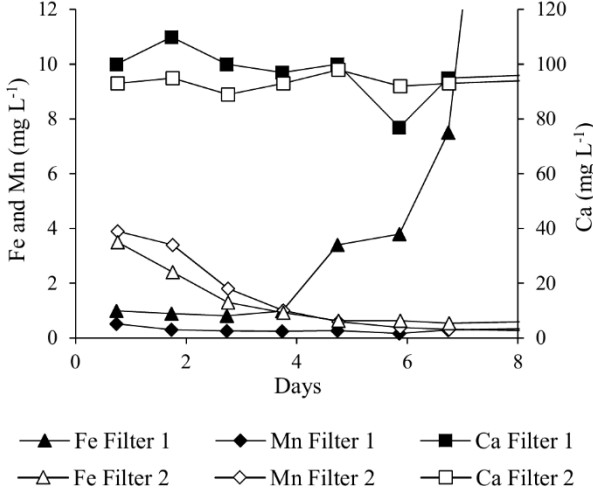

5     **Figure 2: Time series of metal concentrations in backwash water composites.**


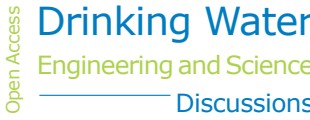

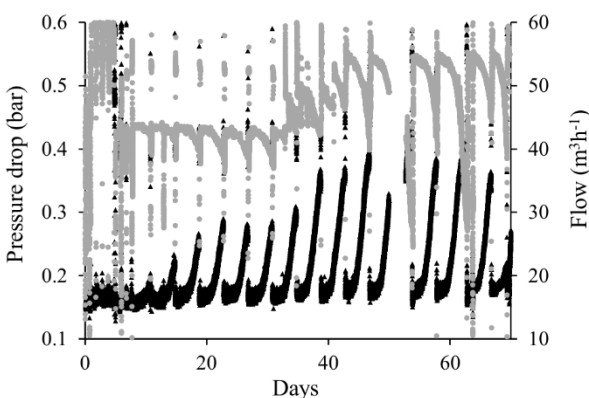

**Figure 3: Time series showing flow and pressure drop over Filter 1.**

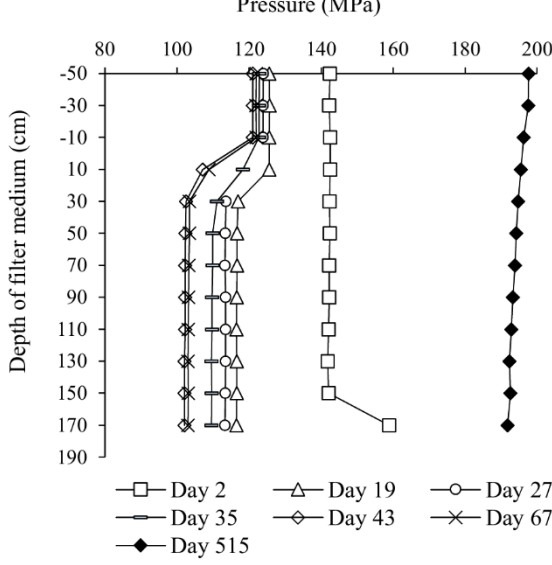

**Figure 4: Pressure profile of Filter 1 over time. Measurements were carried out just prior to a backwash, except Day 2.**

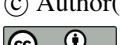

**Figure 5: Time series (top graphs) and depth profiles (bottom graphs) for iron, ammonium and manganese concentrations in water samples from Filter 1.**





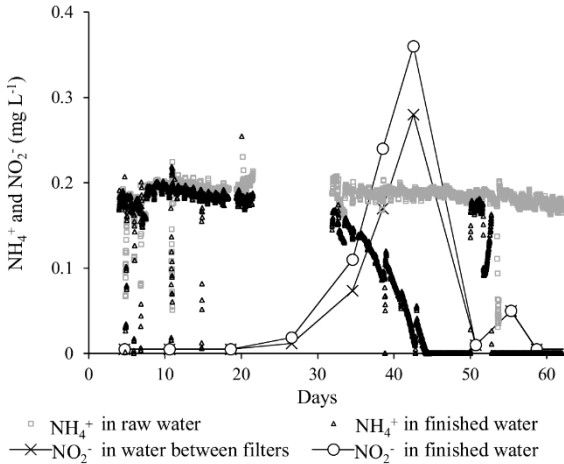

**Figure 6: Time series of ammonium and nitrite concentrations.**

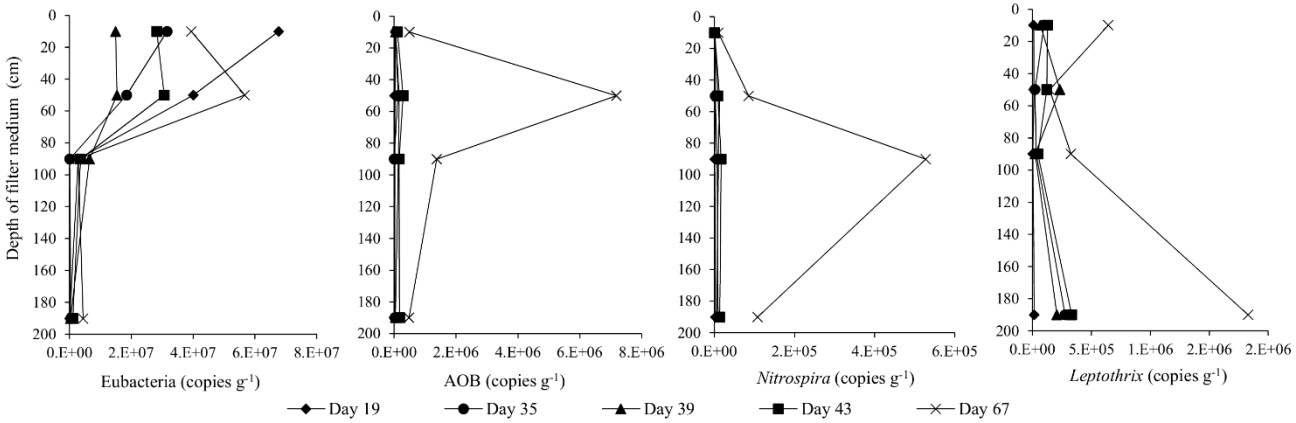

5 **Figure 7: Depth profiles for qPCR results of filter media samples from Filter 1.**



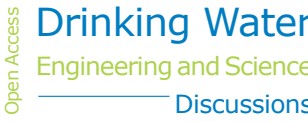

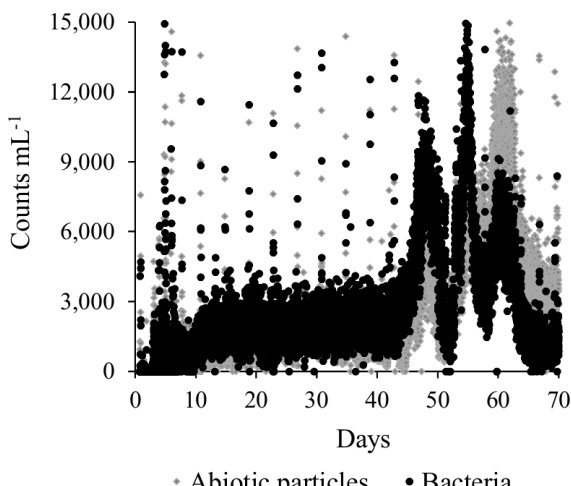

**Figure 8: Time series of bacteria and abiotic particles in finished water.**

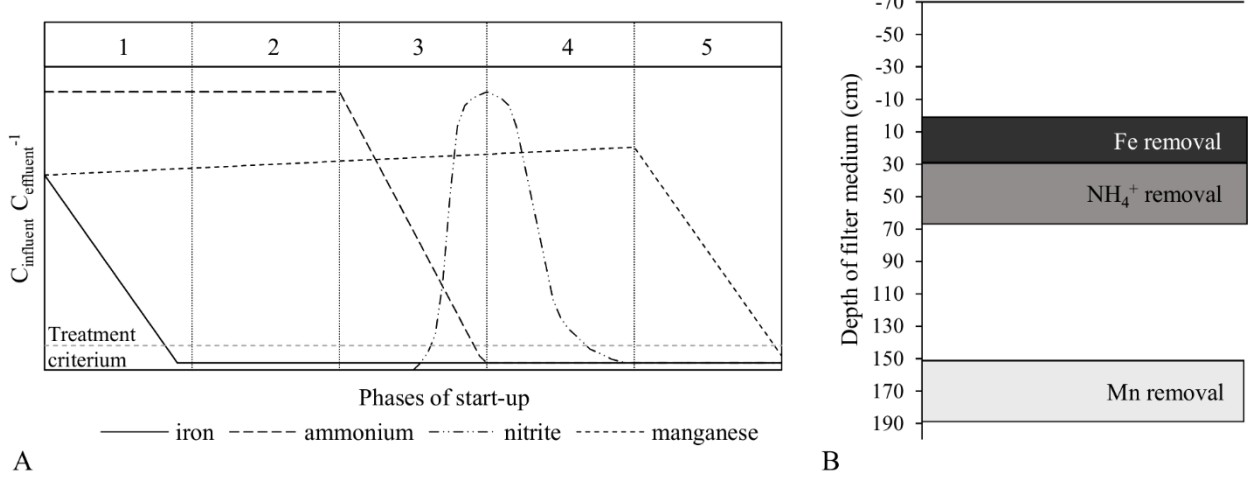

**Figure 9: A. Conceptual phases of the start-up period in drinking water biofilters (modified from Frischherz et al. 1985). B.**

5 **Stratification of iron, ammonium and manganese removal in Filter 1.**