# Peer review of "Comprehensive analysis of the start-up period of a full-scale drinking water biofilter provides guidance for optimization"

_Drinking Water Engineering and Science, 2018_

## Referee Comment (RC1) · Anonymous Referee #1 · 3 Apr 2018

In this paper, the authors successfully start up a practical drinking water treatment line including two biofilters to remove iron, manganese and ammonia from groundwater. A lot of investigates during the start-up period were carried out to better understand the removal process, especially that the related studies about archaea, coliforms and planktonic bacteria seemed to be constructed for the first time. This paper provides guidance for optimization of the biofilter in some degree, however, there are still some questions should be addressed as follows:

1. In the 2.3 section, "the filter backwash was based on a three-step procedure as follows:——-", the backwash strength for this system is stronger than other reports.

[Figure]

Therefore, the authors should clear that if the filtering sand would be washed out of the layer during the start-up period of the system. In addition, it can not see from Fig. 1 that how to drain the backwash wastewater.

2. In the first paragraph of the 3.1 section, "For filter 2, ——- the first 4 days of commissioning". The quartz sand or manganese oxide media are suspect of containing any iron element? It is more likely that the removed iron is the result of chemical contact oxidation of the influent iron.

3. In the 3.5 section, the absence of oxygen that occurred on day 49-51 resulted in the increasing effluent manganese of filter 1 as described. Any filter can intercept solid oxides especially with the relatively smaller quartz sand size of 0.5-0.8 mm in this paper, even if the oxygen deprivation can move the manganese oxide off the sand surface. Therefore, it should be explained that why the manganese could subsequently flow out of the filtering layer. Actually, in absence of dissolved oxygen, the removed manganese (i.e. manganese oxide) may react with some reducing substances (e.g. methane), producing the divalent manganese that can not be trapped by layer. But this reaction will immediately stopped when oxygen is provided, it could not explain why the manganese increased until about 58 days, as the oxygen deprivation only occurred on day 49-51.

4. Generally, there are amounts of manganese oxidizing microbes with a variety of living habits in a bioflter rather than Leptothrix itself. So what is the purpose of this paper use Leptothrix as the sole indicator.

5. The quality of all the figures should be improved.

---

## Author Comment (AC1) · 13 Apr 2018

Dear Anonymous Referee #1

We appreciate the effort by the Anonymous Referee #1 and thank you for your constructive comments to the manuscript. We have provided an individual response to each comment and described in text the changes to the revised final paper.

Answer to comment 1: Small fines in the filter media were washed out with the initial backwashes in the beginning of the start-up period as described in section 3.1. Otherwise, no media is washed out with the backwash water since this leaves the filter in the

same pipe as the inlet water, high above the bed expansion. This will not be added to Figure 1 to avoid confusion with filter inlet, however, to accommodate the comment by the reviewer, the sentence "The inlet pipe is also used for backwash discharge and is located high above the bed expansion" will be added in section 2.3 of the revised final paper.

Answer to comment 2: Filter 2 contains quartz sand (100 % silica) and manganese oxide (which according to the manufacturer can contain up to 3.5 % iron) media. Therefore, iron fines present in the virgin manganese oxide media could be removed with the backwash water. The authors agree that some iron in the backwash water can result from chemical contact oxidation (or rather adsorption and subsequent abiotic oxidation). However, the amount of iron in the media greatly exceeds the iron in the inlet water of Filter 2. Therefore, the initial argumentation in section 3.1 in the discussion paper is retained in the revised final paper.

Answer to comment 3: In section 3.5, there is a reference to a previous study by the same authors that explain why manganese is reduced and dissolved and thereby mobilized out of the filter during the oxygen stop. However, for clarification, the following sentence will be included in the final revised paper: "Manganese oxide reduction and dissolution with concomitant oxidation of Fe(II) has previously been observed in drinking water treatment in conditions of low dissolved oxygen (Bray, R. & Olanczuk-Neyman, K. 2001. The influence of changes in groundwater composition on the efficiency of manganese and ammonia nitrogen removal on mature quartz sand filtering beds. Water Supp. 1 (2), 91–98.)". This means that Fe(II) is the reducing substance and that methane is not needed to explain manganese mobilization. Why manganese was high at Day 58 is not known. Therefore, the following sentence will be added to the revised final paper "The explanation for an additional increase in manganese in effluent water after oxygenation was again working properly is not known".

Answer to comment 4: We agree with the reviewer that Leptothrix is not the only potential MnOB, the genus was targeted with the purpose of iron oxidation. However, as the

genus is known also to have a role in manganese oxidation, presence of this genus in the strata of manganese oxidation is naturally discussed by the authors. Other MnOBs were not targeted due to lack of specific primers for qPCR in the literature.

Answer to comment 5: We agree that the quality of the figures should be improved in the final revised paper, however, these were not requested for the discussion paper. All figures have already been prepared according to the "manuscript preparation" guidelines by Drinking Water Engineering and Science for the final paper.

---

## Referee Comment (RC2) · Anonymous Referee #2 · 1 Jul 2018

The authors present an interesting study focusing on the start-up of the biofilter producing drinking water and this study do provide guidance for practice. It fits the scope of DWES. The amount of data and the novelty reach the level of DWES. The results and discussion section was written clearly. It is good to see that the authors analysed not only the chemical parameters but also the microbial data, besides the microbial results explain the ions removal well. Therefore, I recommend to accept this manuscript after a minor revision. The specific comments are as below.

1. Please ask a native speaker to edit the English because of many language mistakes, such as line 19-20 and 22 in page 1.

[Figure]

2. Too many keywords. Besides, 'start-up' and 'full scale' can stand alone as keywords?

3. In line 5-6 of page 2, the authors mentioned 'this simple treatment train'. I am really interested in it, but i could not find which treatment train. Please add reference here.

4. More literature should be cited in the introduction section.

5. In results, the authors use Fe and iron randomly. Please choose one of them to keep consistent.

---

## Author Comment (AC2) · 5 Jul 2018

Dear Anonymous Referee #2

We thank Anonymous Referee #2 for the effort and valuable comments to the manuscript. We have provided an individual response to each comment and described in text the changes to the revised final manuscript.

Answer to comment 1: We agree with Referee #2 that the language on page 1 and several other places in the article could be improved. After a reading by a native speaker, we have improved the following specific sections in the revised final paper along with

correction of minor mistakes: page 1, lines 19-20; page 1, line 22; page 7, line 29; page 9, line 28; page 12, line 21; page 14, line 15.

Answer to comment 2: We have accommodated this comment by removing the keywords altogether. This decision was based on looking at a number of other articles in DWES in which no keywords were given.

Answer to comment 3: We have added the following sentence to explain the typical treatment train: In Denmark, this simple treatment train is typically composed of e.g. gravity cascades or submerged diffusors to aerate the water followed by rapid sand filtration in open (gravity) or closed (pressure) filters.

Answer to comment 4: We have added three references to the introduction section (page 2, line 30-32).

Answer to comment 5: Based on the comment, we decided to change all references in the text to "iron", "manganese" and "ammonium" while retaining the chemical formulas in all figures and units. This resulted in three changes in the final revised paper (page 8, line 18; page 9, line 26, figure 9A).

All changes are marked with blue in the attached revised paper. Changes from revision after comments from Anonymous Referee #1 are marked in yellow.

Please also note the supplement to this comment:
https://www.drink-water-eng-sci-discuss.net/dwes-2018-6/dwes-2018-6-AC2-supplement.pdf

───────────────────────

**Supplement:**

**Comprehensive analysis of the start-up period of a full-scale drinking water biofilter provides guidance for optimization**

Loren Ramsay[1], Inês L. Breda[1,2], Ditte A. Søborg[1]

[1]Research Group for Energy and Environment, VIA University College, Horsens, 8700, Denmark.

[2]Department of Chemistry and Bioscience, Aalborg University, Aalborg, 9220, Denmark.

*Correspondence to*: Ditte A. Søborg (dans@via.dk)

**Abstract.** The use of biofilters to produce drinking water from anaerobic groundwater is widespread in some European countries. A major disadvantage of biofilters is the long start-up period required for virgin filter medium to become fully functional. Although individual aspects of biofilter start-up have previously been investigated, no comprehensive study in full-scale using inherent inoculation has previously been documented. A thorough investigation of a full-scale drinking water biofilter was carried out over 10 weeks of start-up. The many spatial and temporal changes taking place during start-up were documented using a holistic approach. In addition to collection of many samples over time (frequency) and space (filter depth), this study entailed the use of multiple sample media (water, backwash water and filter media) and multiple types of analyses (physical, chemical and microbiological). The decrease in filter effluent concentrations of individual substances to compliance levels followed a specific order that was shown to coincide with the spatial-temporal development of bacteria on the filter media. Due to the abiotic nature of the iron removal process, iron disappears earliest in the start-up period followed by substances that require growth of microorganisms. Ammonium disappears next, with nitrite appearing briefly near the end of ammonium removal, followed by manganese. The thorough overall picture obtained by these efforts provides guidance for optimization and monitoring of the start-up. Guidance for optimization includes shortening the start-up by focusing on kick-start of the ammonium removal, limiting the monitoring burden (at-line measurements of ammonium in finished water supplemented with manual manganese measurements when ammonium removal is complete), and improving filter design by isolating the removal processes in separate, smaller filters.

**1 Introduction**

Biofilters for the production of drinking water from anaerobic groundwater have been in widespread use for decades in some European countries and are currently gaining interest in North America. In this paper, biofilters are defined as submerged, granular, rapid filters in which the individual grains of the filter media have developed a natural coating capable of performing the desired treatment. A typical waterworks may be composed of an aeration step followed by biofiltration. In Denmark, this simple treatment train is typically composed of e.g. gravity cascades or submerged diffusors to aerate the water followed by rapid sand filtration in open (gravity) or closed (pressure) filters. These processes are 
[revised manuscript text omitted]

Zhu, J.; Tang, X.; Wu, Z.; Hen, H. (2018) Migration and Control of Invertebrates in Waterworks with Advanced Treatment. J. Environmental Engineering (United States) 144(7).

**Table 1: Raw water quality at Truelsbjerg waterworks.**

| Parameter | Unit | Average | Std. Dev. | Danish criteria[b] | No. of samples |
|---|---|---|---|---|---|
| **Treatment substances** | | | | | |
| Oxygen[a] | mgL$^{-1}$ | 0.22 | 0.12 | >5 | 9 |
| Iron | mgL$^{-1}$ | 1.40 | 0.19 | 0.1 | 26 |
| Manganese | mgL$^{-1}$ | 0.45 | 0.07 | 0.02 | 28 |
| Ammonium | mgL$^{-1}$ | 0.21 | 0.04 | 0.05 | 26 |
| Methane | mgL$^{-1}$ | 0.07 | 0.02 | 0.01 | 3 |
| **Major ions** | | | | | |
| Calcium | mgL$^{-1}$ | 89 | 5 | 200 | 4 |
| Magnesium | mgL$^{-1}$ | 7.6 | 0.3 | 50 | 3 |
| Sodium | mgL$^{-1}$ | 15 | 0.0 | 175 | 3 |
| Potassium | mgL$^{-1}$ | 2.2 | 0.0 | 10 | 3 |
| Hydrogen carbonate | mgL$^{-1}$ | 282 | 4 | >100 | 4 |
| Chloride | mgL$^{-1}$ | 25 | 1 | 250 | 9 |
| Sulfate | mgL$^{-1}$ | 38 | 5 | 250 | 9 |
| Nitrate (as $NO_3^-$) | mgL$^{-1}$ | <0.5 | 0.0 | 50 | 8 |
| **Bacteria** | | | | | |
| HPC, 22°C | mL$^{-1}$ | 17 | 11 | 50 | 8 |
| HPC, 37°C | mL$^{-1}$ | 1.9 | 1.5 | 5 | 8 |
| Coliforms | 100mL$^{-1}$ | <1 | 0.0 | <1 | 8 |
| *E. coli* | 100mL$^{-1}$ | <1 | 0.0 | <1 | 8 |
| **Others** | | | | | |
| Nitrite (as $NO_2^-$) | mgL$^{-1}$ | 0.01 | 0.01 | 0.01 | 17 |
| Phosphorous (as P) | mgL$^{-1}$ | 0.08 | 0.02 | 0.15 | 4 |
| NVOC (as C) | mgL$^{-1}$ | 1.6 | 0.1 | 4 | 9 |
| Hydrogen sulfide[c] | mgL$^{-1}$ | <0.02 | 0.00 | 0.05 | 3 |
| pH[a] | | 7.3 | 0.06 | 7.0-8.5 | 9 |
| Conductivity[a] | mSm$^{-1}$ | 58 | 0.8 | >30 | 9 |
| Temperature[a] | °C | 8.9 | 0.1 | 12 | 9 |

Shading indicates concentrations above Danish drinking water criteria.

[a] Field measurements.

[b] Criteria for water exiting the waterworks (BEK 802, 2016).

[c] Very faint hydrogen sulfide odor is detectable at the waterworks.

[Figure]

Figure 1: Initial design of a production line at Truelsbjerg waterworks (revised from Søborg et al. 2015).

[Figure]

5    Figure 2: Time series of metal concentrations in backwash water composites.

[Figure]

**Figure 3: Time series showing flow and pressure drop over Filter 1.**

[Figure]

**Figure 4: Pressure profile of Filter 1 over time. Measurements were carried out just prior to a backwash, except Day 2.**

[Figure]

**Figure 5: Time series (top graphs) and depth profiles (bottom graphs) for iron, ammonium and manganese concentrations in water samples from Filter 1.**

[Figure]

**Figure 6: Time series of ammonium and nitrite concentrations.**

[Figure]

5    **Figure 7: Depth profiles for qPCR results of filter media samples from Filter 1.**

[Figure]

**Figure 8: Time series of bacteria and abiotic particles in finished water.**

[Figure]

A                                                                        B

**Figure 9: A. Conceptual phases of the start-up period in drinking water biofilters (modified from Frischherz et al. 1985). B.**
5 **Stratification of iron, ammonium and manganese removal in Filter 1.**